# Experimental and Numerical Validation of Whispering Gallery Resonators as Optical Temperature Sensors

**DOI:** 10.3390/s22207831

**Published:** 2022-10-15

**Authors:** Franzette Paz-Buclatin, Ylenia Perera-Suárez, Inocencio R. Martín, Susana Ríos, Omar de Varona, Airán Ródenas, Leopoldo L. Martin

**Affiliations:** 1Departamento de Física, Universidad de La Laguna, Apdo. 456, E-38200 San Cristóbal de La Laguna, Spain; 2Instituto Universitario de Estudios Avanzados (IUdEA), Universidad de La Laguna, Apdo. 456, E-38200 San Cristóbal de La Laguna, Spain

**Keywords:** temperature sensor, whispering gallery modes, laser heating, microsphere

## Abstract

This study experimentally and numerically validates the commonly employed technique of laser-induced heating of a material in optical temperature sensing studies. Furthermore, the Er3+-doped glass microspheres studied in this work can be employed as remote optical temperature sensors. Laser-induced self-heating is a useful technique commonly employed in optical temperature sensing research when two temperature-dependent parameters can be correlated, such as in fluorescence intensity ratio vs. interferometric calibration, allowing straightforward sensor characterization. A frequent assumption in such experiments is that thermal homogeneity within the sensor volume, that is, a sound hypothesis when dealing with small volume to surface area ratio devices such as microresonators, but has never been validated. In order to address this issue, we performed a series of experiments and simulations on a microsphere supporting whispering gallery mode resonances, laser heating it at ambient pressure and medium vacuum while tracking the resonance wavelength shift and comparing it to the shift rate observed in a thermal bath. The simulations were done starting only from the material properties of the bulk glass to simulate the physical phenomena of laser heating and resonance of the microsphere glass. Despite the simplicity of the model, both measurements and simulations are in good agreement with a highly homogeneous temperature within the resonator, thus validating the laser heating technique.

## 1. Introduction

Optical sensors based on microresonators supporting Whispering Gallery Mode (WGM) resonances have been gaining interest for some time due to their exceptional sensitivity, structural diversity, and ease of integration with existing infrastructures [1]. WGM are morphology-dependent electromagnetic resonances produced from the multiple total internal reflections of the light travelling along the curved surface of a resonator. Such resonances are commonly supported in circularly symmetric morphologies such as microspheres [2], microtoroids [3], and microbottles [4], and also in asymmetric resonant cavities like a rounded-isosceles-triangle-shaped microcavity [5]. Typically, WGM resonances are observed through evanescent coupling of light into the microresonator with the use of a prism [6], a tapered fiber [7], or an angle-polished fiber [8]. A simpler approach that does not require a coupler would be doping the microresonator with a fluorescent substance and taking advantage of the Purcell enhancement effect, which produces a notable increase in the emission intensity at the resonant wavelengths of the microresonator [9].

In a geometrical approximation valid for a resonator with radius a >> λ, the ray circles the cavity through multiple total internal reflections and returns in phase such that the following resonance condition is observed [10]:2πan = lλ(1)
where n is the refractive index of the resonator, λ is the resonant wavelength, and l is the polar mode number. Therefore, a small change in the size and/or refractive index of the microresonator can cause a significant shift in the resonant wavelength for a given mode. Meanwhile, the material properties of a resonator, such as its size and refractive index, are susceptible to thermal fluctuations according to its thermal expansion α and thermo-optic coefficient β, respectively [11]. This is the main principle of WGM-based temperature sensors, where a temperature change ΔT is directly related to a resonant wavelength shift Δλ according to the following equation:Δλ = (α + β)λΔT(2)

In most studies regarding WGM-based temperature sensors, the temperature change is induced either through the variation of the ambient temperature [11,12] or through the absorption of the laser pump energy [13,14]. In this latter method, there is an issue of non-homogenous heating of the microsphere due to the laser heating nature. This produces a temperature gradient from higher to lower temperatures from the center of the microsphere to its surface, where the WGM are located. In this work, we aim to study the feasibility of measuring the temperature at the surface of a laser-heated microresonator using the WGM, despite the issue of non-homogenous heating. This involves studying the WGM displacement with the temperature change induced through both methods: ambient temperature variation and laser heating at ambient pressure and medium vacuum as a way to thermally isolate the microsphere. We complement our experimental results with a COMSOL^®^ heat transfer simulation of the laser heating.

## 2. Materials and Methods

The microresonator employed in this study is an Er^3+^-doped Barium Titanium Silicate (BTS) microsphere. The bulk glass with a composition of 40% BaO, 20% TiO_2_, 40% SiO_2_ and doped in excess with 3% Er_2_O_3_ (in molar ratio) was produced through the conventional melt-quenching method of the precursor components at 1500 °C for one hour. After that, the microspheres were produced by dropping a crushed portion of the obtained glass through a small torch. The crushed glass melts as it passes through the flame, then surface tension pulls it into a sphere which quenches into an amorphous state as it drops to the cooler region outside the hot zone [15]. A microsphere with a radius of 10 µm was selected for the experiment.

A modified confocal microscope setup, shown in Figure 1, was used in order to observe the WGM. A continuous 532 nm DPSSL was used to excite the Er^3+^-doped microsphere attached to a stem inside a vacuum chamber. The luminescent emission is observed through a CCD spectrograph. The laser beam was focused on the center of the microsphere while the luminescent emission at its surface was collected so that the highest WGM resonance amplitudes were observed [16]. In order to characterize the microsphere as a temperature sensor, it was heated in two different ways: by increasing the laser pump power and by increasing the surrounding temperature of the sample through a water bath. Furthermore, in order to study the heating mechanism of the microsphere, the experiments were done in two conditions: at atmospheric pressure and at 20 Pa (medium vacuum).

## 3. Results and Discussion

### 3.1. Emission Spectrum of Er^3+^

The WGM resonances superimposed with the typical Er^3+^ emission spectrum are shown in Figure 2 as a result of the 532 nm excitation of the microsphere. The emission bands of Er^3+^ are observed at 650 nm (^4^F_9/2_ → ^4^I_15/2_), 800 nm (^2^H_11/2_ → ^4^I_13/2_), 850 nm (^4^S_3/2_ → ^4^I_13/2_), and 1000 nm (^4^I_11/2_ → ^4^I_15/2_). The WGM at 670 nm, 850 nm, and 1000 nm bands were used for the temperature correlation.

### 3.2. Temperature Correlation with WGM Displacement

#### 3.2.1. Laser Heating

To obtain the temperature correlation by laser heating, the laser pump power was increased from 2 mW to 112 mW and the WGM displacement from the three mentioned emission bands was recorded. This is shown in Figure 3 for the microsphere at atmospheric pressure (a) and at 20 Pa vacuum (b). A linear tendency can be observed for each correlation. The displacement rates obtained at atmospheric pressure and in a vacuum are displayed in Table 1. It can be seen that the displacement rate depends on the wavelength, as expected from Equation (2). It can also be seen from these results that the displacement rates of the microsphere in a vacuum are greater than those of the microsphere at atmospheric pressure. This is because, in vacuum, heat transfer is greatly reduced, allowing only radiation (from the pump laser to the microsphere as a heating mechanism and from the microsphere to the surrounding medium as a cooling mechanism) and conduction via the microsphere stem. A slight saturation can also be observed for higher pump powers in the vacuum experiment. This could be due to the higher radiation loss from the microsphere as its temperature rises, which opposes the laser heating.

#### 3.2.2. Water Bath Heating

For the water bath heating, the experiment was done at atmospheric pressure and the temperature of the microsphere was heated from 298 K to 328 K. The WGM displacement of the 850 nm band as a function of temperature is shown in Figure 4 (blue). With a linear fit, a displacement rate of 6.6 ± 0.8 pm/K was obtained.

### 3.3. Simulations of the WGM Displacement and Laser Heating of the Microsphere

In order to obtain the relationship between the temperature of the microsphere and the laser pump power, two simulations were done with COMSOL Multiphysics^®^: a simulation of the displacement of the WGM frequencies of the microsphere with temperature using the Frequency Domain (ewfd) solver and a simulation of the laser heating of the microsphere using the Heat Transfer in Solids solver.

#### 3.3.1. WGM Displacement with Temperature Simulation

The WGM in a microsphere was simulated using the Frequency Domain (ewfd) solver included in the Wave Optics package. This solves the eigenfrequencies of the resonator modes numerically. To reduce the computation time, an axisymmetric 2D geometry was implemented. The initial radius of the microsphere is set to 10 µm, similar to the one used in the experiment. To properly simulate the WGM shift of the microsphere with temperature, its refractive index, thermal expansion coefficient, and thermo-optic coefficient were previously determined. The refractive index of the BTS glass was found to be 1.76 at 633 nm for a BTS glass of similar composition in Ref. [17]. Meanwhile, the thermal expansion coefficient was measured with a dilatometer while the thermo-optic coefficient was measured using a Michelson interferometer as detailed in Ref. [18], and values of 7.94 × 10^−6^ K^−1^ and 1.63 × 10^−6^ K^−1^ were obtained, respectively. The profile of the electric field norm of a WGM in the microsphere is shown in Figure 5, and the simulated WGM shift of the 850 nm band with temperature is shown in Figure 4 (red). A displacement rate of 8.04 ± 0.37 pm/K was obtained from the simulation, similar to the 6.57 ± 0.32 pm/K experimentally obtained previously in Section 3.2.2.

#### 3.3.2. Laser Heating of the Microsphere Simulation

To model the laser heating experiment, the Heat Transfer in Solids solver interface was employed. This simulates the temperature rise within and around a material. The model is shown in Figure 6a, which consists of the following: a sphere of a 10 µm radius that represents the BTS microsphere, a smaller sphere of a 2 µm radius acting as a heat source at the center of the first sphere that represents the laser spot volume, a cylindrical steel bar representing the needle that supports the microsphere, and surrounding all of these elements is a box of air with a side of 2 mm. In this model, we set the steel bar at constant room temperature since one end of it is in constant contact with the wall of the vacuum chamber, which remains at room temperature, and due to its large size and high conductivity, the heat from the microsphere is negligible according to the simulations. A 500 nm gap between the bar and the microsphere was added in order to account for the roughness of the needle. We assume that all the energy absorbed by the microsphere is converted into heat. Furthermore, we neglect the laser absorption in the whole path of the laser within the microsphere and only consider the absorption in the focus since the energy density here is much greater than out of focus. The absorption coefficient of the microsphere at 532 nm is also a required parameter for this simulation and is measured to be 5.69 cm^−1^. Aside from this, we took into consideration the losses from having the laser beam pass through a sapphire window in the vacuum chamber as well as the reflection losses from the microsphere, such that the simulated microsphere absorbs about 0.3% of the incident laser power. The thermal conductivity of air at atmospheric pressure and 20 Pa was obtained from Ref. [19] with values of 2.50 × 10^−2^ W/mK and 4.39 × 10^−3^ W/mK, respectively.

The temperature map of the microsphere and its surroundings when laser heated with 112 mW is shown in Figure 6b,c for atmospheric pressure and vacuum, respectively. To quantify the inhomogeneity of the temperature distribution, the following figure of merit is defined:(3)$$\%\mathrm{Variation}=(T_{\mathrm{center}}-T_{\mathrm{surface}})/T_{\mathrm{center}}$$
where T_center_ and T_surface_ are the temperatures at the center and at the surface of the microsphere, respectively. For atmospheric pressure simulations, the temperature varies within the microsphere by 2.1%, while for vacuum simulations, this value is 1.3%. As can be seen, although the microsphere is heated at its center, there is high homogeneity in the temperature distribution throughout the microsphere, which verifies temperature measurements on its surface through WGM displacements. Furthermore, such homogeneity is enhanced in vacuum experiments because of the lessened interaction between the surface of the microsphere and its surroundings. 

By changing the power of the heat source and recording the change in temperature at the surface of the microsphere produced by the simulation, the relationship between the laser pump power and temperature change can be obtained. The simulated data points for both atmospheric pressure and vacuum configurations are shown in Figure 7. Meanwhile, the experimental data points were obtained by combining the correlations established in Figure 3 (850 nm band) and Figure 4. As can be seen, the simulations reproduce the experimental results, and a linear dependence exists between the temperature and laser pump power. With the microsphere at atmospheric pressure, laser heating rates of 0.42 K/mW and 0.39 K/mW were obtained experimentally and numerically, respectively. With the microsphere in vacuum, rates of 1.97 K/mW and 2.25 K/mW were obtained experimentally and numerically, respectively. As can be seen, the laser heats the microsphere much more easily in a vacuum than in air for the reason explained previously. Moreover, the laser heating rates in the simulation are similar to those in the experiment, both for air and vacuum. This further supports the validity of such temperature measurements using WGM displacements and laser-originated heating. The noticeable difference between the simulation and experimental data in vacuum can be attributed to the high uncertainty in the measurement of the exact pressure of the vacuum since the vacuum chamber had leaks.

## 4. Conclusions

In this study, we have developed a WGM-based temperature sensor from Er^3+^-doped BTS microspheres. We established a WGM displacement—pump power correlation through laser heating of the microsphere and obtained displacement rates of 2.8 pm/mW and 13.0 pm/mW for the 850 nm emission band at room pressure and at a 20 Pa vacuum, respectively. On the other hand, we established a WGM displacement—temperature correlation through water bath heating and obtained a displacement rate of 6.57 pm/K for the same emission band. In order to verify if one can heat the microsphere from its center (such as in laser heating) and measure the temperature through the WGM near its surface, we performed a laser heating simulation through COMSOL. The simulated and experimentally obtained temperature—pump power correlations are very similar to each other both for the microsphere at room pressure and in vacuum. This suggests that the proposed method of using the WGM to study how much the laser heats up the microsphere makes sense. Furthermore, laser heating rates of 0.42 K/mW and 1.97 K/mW were obtained for the microsphere in air and in vacuum, respectively. This shows that it is much easier to heat the microsphere in vacuum, which can be attributed to the fewer heat transfer mechanisms from the microsphere in vacuum.

## Figures and Tables

**Figure 1 sensors-22-07831-f001:**
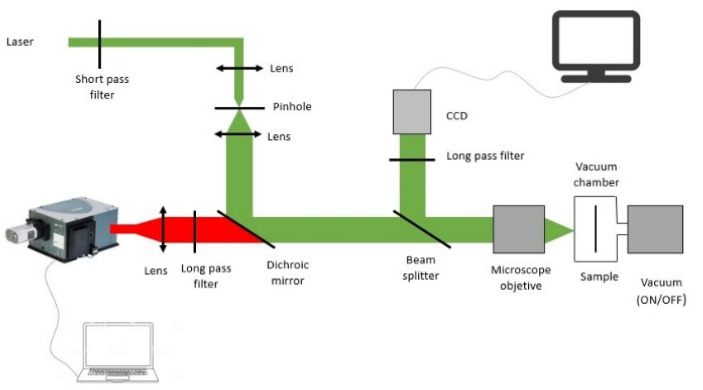
Modified confocal microscope setup used to observe the WGM.

**Figure 2 sensors-22-07831-f002:**
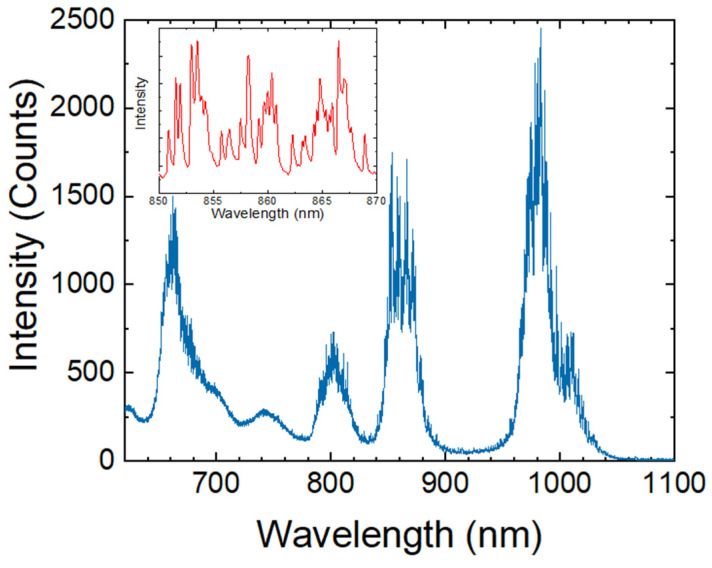
Emission spectrum of Er^3+^-doped BTS glass microsphere under a 532 nm excitation. Inset: Close up of the WGM resonant peaks in the 850 nm emission band.

**Figure 3 sensors-22-07831-f003:**
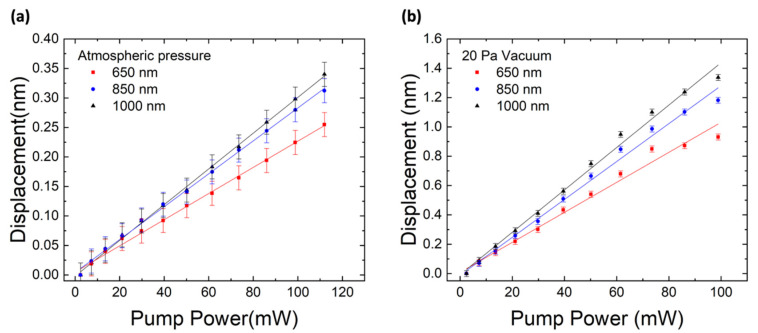
Displacement of the WGM peaks as a function of laser pump power (**a**) at atmospheric pressure and (**b**) at 20 Pa vacuum. Error bars represent standard error of 20.4 pm.

**Figure 4 sensors-22-07831-f004:**
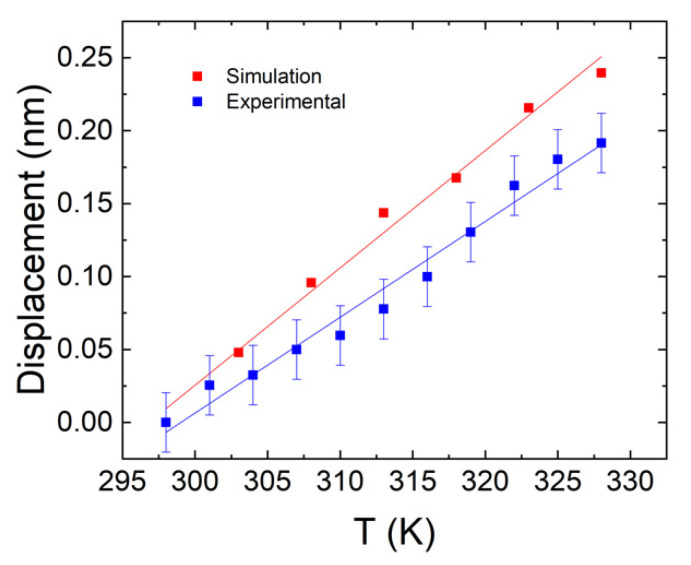
Displacement of the WGM peaks (at 850 nm) as a function of temperature, obtained from experimental data (blue) and simulations (red). The error bars represent the standard error of 20.4 pm.

**Figure 5 sensors-22-07831-f005:**
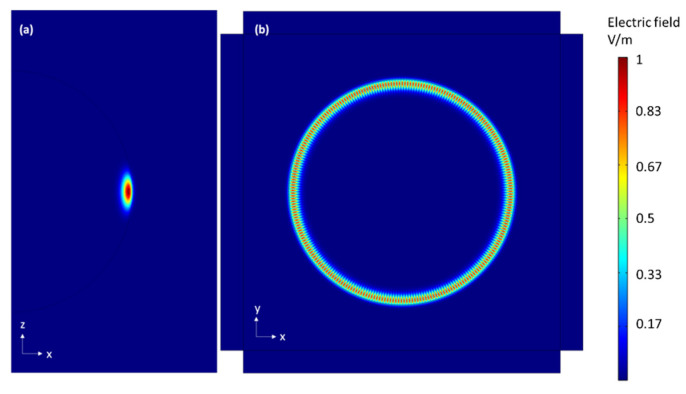
Electric field norm profile of a WGM with TE polarization with n = 1, m = l = 133, λ=826.63 nm with (**a**) z axisymmetric simulation and (**b**) transversal simulation. Color represents electric field intensity normalized in rainbow scale.

**Figure 6 sensors-22-07831-f006:**
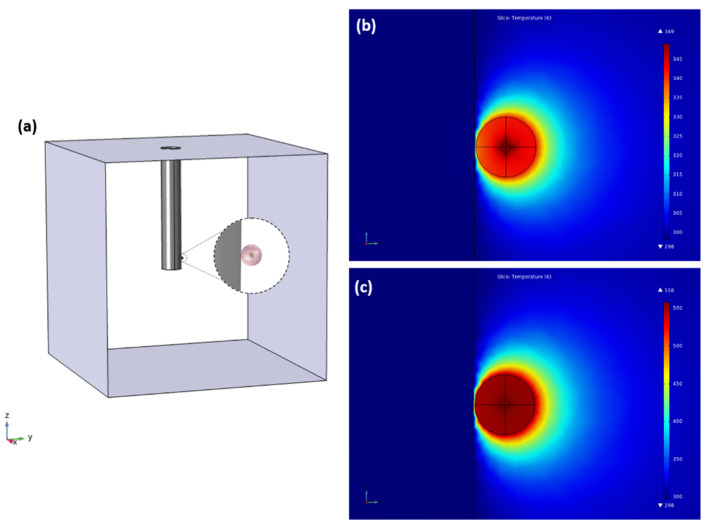
(**a**) Geometry of the laser heating simulation. Temperature map in (K) within and around the microsphere heated with 112 mW of pump power at (**b**) atmospheric pressure and (**c**) 20 Pa vacuum.

**Figure 7 sensors-22-07831-f007:**
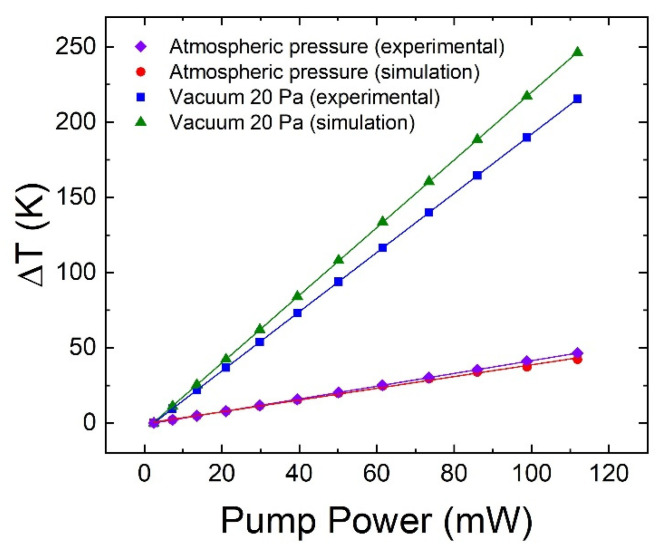
Simulated and measured changes in temperature as a function of laser pump power for the microsphere at atmospheric pressure and at 20 Pa vacuum.

**Table 1 sensors-22-07831-t001:** Displacement rates obtained as a function of laser pump power.

Emission Band	Displacement Rates (pm/mW)	
	at Atmospheric Pressure	at 20 Pa Vacuum
650 nm	2.22±0.16	10.2±0.5
850 nm	2.79±0.16	12.9±0.4
1000 nm	3.03±0.16	14.5±0.4

## Data Availability

The data presented in this study are available upon request from the corresponding author.

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
