# Peer review of "Experimental and Numerical Validation of Whispering Gallery Resonators as Optical Temperature Sensors"

_sensors, 2022, doi:10.3390/s22207831_

Round 1

Reviewer 1 Report

The paper by F. Paz-Buclatin et al. considers the dielectric microspheres for using as optical temperature sensors. Specifically, the authors investigate the WGM excitation in BTS Er3+-doped microspheres upon intrinsic optical absorption and ambient temperature changes. The corresponding experimental and numerical simulations results on WGM resonance shift in atmospheric pressure and vacuum conditions are presented. Mostly, the results reported in the paper are straightforward and previously reported elsewhere. The certain novelty of the paper under review consists in the treatment of the heat transfer problem in the microsphere locally heated by a focused laser beam. Although from general considerations, the temperature distributions obtained are trivial and expected.

The paper is interesting, possesses the logic of reasoning, and the findings are sufficiently illustrated. However, before considering for publication the paper by F. Paz-Buclatin et al. demands major revision.

Comments.

1.      Introduction, second sentence: Note that WGMs are observed not only in “a circularly symmetric” particle, but also can be excited in shaped microcavities such as, e.g., triangle (Laser Photonics Rev. 10, 40: 2016) or rectangle structures (DOI: 10.1002/andp.201900033).

2.      Indicate that eq.1 is an approximate formula derived in the geometric optics limit.

3.      Section 3. What is the meaning of the inset in fig.2? Why an 850nm emission band is detailed?

4.      Experimental points in figs. 3 and 4 lacks STD bars as presented in Table 1.

5.      Table 1 is confusing because the column header “Displacement…” does not correspond to the emission bands values.

6.      Which is the resonant wavelength of TE_133 WGM plotted in fig 5? Does it correspond to the BTS microsphere refractive index measurements at 633nm (1.76)?

7.      In fig.5, designate the symmetry axis, left and bottom axes also.

8.      Section 3.3.2. The relative position of a small sphere (2um) simulating the laser heating volume is unclear. Is it located on the illuminated or shadow hemisphere of a bigger BTS sphere (10um)? Worthwhile noting, in optically big spheres (2*pi*lam/a >>1) exposed to a laser radiation the internal optical field intensity is maximal near a shadow hemisphere.

9.      In general, I wonder why the authors did not use the Multiphysics feature of COMSOL software when simulating the laser heating of a microsphere? In my opinion, this could be done in a 2D-geometry especially since the stationary COMSOL solver is exploited.

10.  The authors note below fig. 6 that “such homogeneity is enhanced in vacuum experiments…”. How the temperature distribution homogeneity is assessed? According to which merits?

Author Response

The authors would like to thank the reviewers for all useful and helpful comments on our manuscript. All comments have been taken into account and the paper has been revised accordingly.

Reviewer 2 Report

The authors performed a series of experiments and simulations on an Er3+-doped BTS microsphere, laser heating it at different pressures while tracking the resonances wavelength shift and comparing it to the shift rate observed in a thermal bath. It’s a topic of interest to the researchers in the related areas. The manuscript was clearly written. I think this is a high-quality paper and should be accepted with just minor revisions for publication in MDPI sensors.

1.     Add several relevant references to the introduction.

2.     In paragraph of 3.3.1, 2D geometry was implemented, the authors should better give some 3D simulations results for comparison.

Author Response

(The authors gave the same response as above.)

Round 2

Reviewer 1 Report

The authors have properly addressed all my comments and suggestions. The paper can be published in the present form.